# Wound Dressing Double-Crosslinked Quick Self-Healing Hydrogel Based on Carboxymethyl Chitosan and Modified Nanocellulose

**DOI:** 10.3390/polym15163389

**Published:** 2023-08-13

**Authors:** Anshan Huang, Yehong Chen, Chaojun Wu

**Affiliations:** State Key Laboratory of Biobased Material and Green Papermaking, Qilu University of Technology (Shandong Academy of Sciences), Jinan 250353, China; huanganshan98@163.com

**Keywords:** cellulose nanofibrils, cellulose nanocrystals, self-healing hydrogel

## Abstract

The use of hydrogels in wound dressings, which is pivotal for effective wound treatment, has been widely applied to diverse medical wound conditions. However, formulating natural hydrogels that combine robust strength and self-healing capabilities is a significant challenge. To overcome this, we successfully designed a natural nanocellulose self-healing hydrogel that can quickly self-heal and restore the complete hydrogel structure after injury to fill the injured area and protect the wound from external damage. Our study utilized modified natural polymer carboxymethyl chitosan (CMC), hydrazide-modified carboxymethyl cellulose nanofibers (HCNF), and cellulose nanocrystals modified by dialdehyde (DACNC) to fabricate the hydrogel. The amides containing more amino groups and HCNF in CMC can be used as cross-linking nodes, and the high aspect ratio and specific surface area of DACNC are favorable for the connection of many active hydrogels. The hydrogel is crosslinked by the dynamic imide bond and hydrazone bond between the amino group of CMC, the amide of HCNF, and the aldehyde of DACNC and has a double network structure. These connections can be readily reassembled when disrupted, enabling fast self-healing of hydrogels within five minutes. Moreover, HCNF and DACNC were incorporated as nano-reinforced fillers to bolster the hydrogel’s strength while preserving its high liquid absorption capacity (381% equilibrium swelling rate).

## 1. Introduction

The skin is an essential organ that envelops the body and interacts with the external environment, offering sensory input, temperature regulation, and protection from external threats [1,2]. Skin trauma, originating from mechanical abrasions, burns, or surgical procedures, triggers a complex and dynamic tissue regeneration and growth process encompassing four stages: coagulation and hemostasis, inflammation, tissue proliferation, and remodeling [3,4]. This process incurs pain and produces substantial tissue fluid due to surface tension and capillary forces. To avert wound infection and stimulate healing, skin dressings must fulfill several criteria, including histocompatibility, moisturizing capabilities, physical and mechanical strength, surface microstructure, and biochemical properties, to encourage cell adhesion and differentiation [5]. Traditional wound gauzes used for drying wounds can absorb exudates. However, frequent replacement is necessary to prevent impregnation and adherence to the wound, which could precipitate wound infection. Hydrogels, with their 3D network structure crosslinked by physics, chemistry, or polymerization, offer several benefits as wound repair materials. They shield the wound and provide a moist environment for repair [6,7,8]. Notably, natural polymer hydrogels exhibit promising application prospects in biomaterials owing to their excellent biocompatibility and biodegradability.

Numerous natural polymer hydrogels, including chitosan [9], alginate [10], and cellulose [11], have been employed in wound dressings to expedite wound healing. However, these natural hydrogels tend to be fragile and readily disintegrate under external mechanical stress, posing a risk of wound deterioration due to external bacterial invasion. Self-healing hydrogels can not only augment the mechanical strength of the hydrogel through nano-reinforced materials but also confer the hydrogel with self-healing ability through dynamic bonding, repair its structural damage post-stress, and provide continual protection for skin wound healing [12,13]. Merging nano-reinforced materials and dynamic cross-linking appears to be a viable approach for developing self-healing natural hydrogels with superior strength for wound dressings. For instance, Huang et al. synthesized aldehyde-modified cellulose nanocrystals and carboxymethyl chitosan to produce self-healing hydrogels via the Schiff base reaction. The presence of dynamic imine bonds allows the hydrogel to restore the gel network [14]. Li et al. utilized modified dialdehyde bacterial cellulose (DABC) and chitosan as raw materials to form a hydrogel by imine bond [15]. Ying et al. prepared self-healing hydrogels using the reaction between the aldehyde and amino groups of the dialdehyde carboxymethyl cellulose and propionyl hydrazide, adjusting their mechanical strength by adding cellulose nanofibers (CNF) [16]. All these studies added reinforcements in isolation, leading us to consider whether nanomaterials as reinforcements could also act as amino donors to enhance the self-healing ability.

To test this hypothesis, we designed a self-healing nanocomposite natural hydrogel. Carboxymethyl cellulose nanofibers (CMCNF) were interconnected with hydrazide, cellulose nanocrystals modified with aldehydes, and carboxymethyl chitosan through the Schiff reaction. Chitosan, a renewable natural polysaccharide compound extracted from shell organisms, is extensively used in medicine due to its inherent biodegradability and good biocompatibility [17,18]. However, owing to poor water solubility, chitosan can only dissolve in acidic environments. Carboxymethyl chitosan modified with a hydrophilic group is soluble in water. The hydrogel made from carboxymethyl chitosan (CMC) exhibits good biocompatibility and moisturizing capabilities [19]. Similar to chitosan, carboxymethyl chitosan is rich in primary amine groups. Its amine group can react with aldehyde groups to synthesize dynamic reversible imine bonds, which are easy to break and reshape, thereby enabling the hydrogel to repair quickly. The self-healing function of the hydrogel is attributed to the reversible chemical bonds in its network.

CNF are natural polysaccharides with a natural crystal structure renowned for their good mechanical properties, renewability, biocompatibility, biodegradability, and large surface [20]. Cellulose nanocrystals, which are natural biopolymers extracted from renewable wood, are biocompatible and suitable for biomedicine [21,22]. As nanomaterials, they have a high specific surface area and excellent mechanical strength [23,24]. These attributes make carbon nanocrystals suitable as reinforced nanomembranes in polymer hydrogel matrices. It has been reported that the crystal structure of aldehyde-modified cellulose nanocrystals remains unchanged post-oxidation [25], and the introduced aldehyde group can act as a cross-linking site to chemically cross-link with some polymer chains, allowing cellulose nanocrystals (CNC) to serve as a reinforcing substance; CNC can cross-link as a reactant through aldehyde group modification to enhance the effect. Given that CMC is rich in amino groups, cellulose nanocrystals modified by dialdehyde (DACNC) have a high specific surface area and an exposed aldehyde group, and DACNC can enhance the network structure in hydrogels. We hypothesized that a self-healing hydrogel composed of hydrazide-modified nanofibers, carboxymethyl cellulose nanofibers (HCNF), DACNC, and CMC can rapidly form a gel under physiological conditions and heal swiftly when ruptured. When the hydrogel is subjected to physical action, DACNC, with its high specific surface area and active aldehyde group, is used in conjunction with HCNF to strengthen and protect the network structure of hydrogels.

Considering these factors, we designed a nanocomposite natural hydrogel with self-healing capabilities based on hydrazide and imide bonds. The hydrogel consists of HCNF, CMC, and DACNC. The self-healing of the hydrogel is attributed to the dynamic imide and hydrazide bonds. In this study, we explored the chemical properties, surface morphology, and self-healing properties of these hydrogels.

## 2. Experimental

### 2.1. Materials

CMCNF and CNC were sourced from Tianjin Wood Elf Biological Technology Co., Ltd. Water-soluble carboxymethyl chitosan, sodium periodate, polyethylene glycol, sodium hydroxide, adipic dihydrazide, *N*-(3-dimethylaminopropyl)-*N*’-ethylcarbodiimide (EDC), and 1-hydroxybenzotriazole (HOBt) were obtained from Shanghai Macklin Biochemical Co., Ltd. (Shanghai, China). Mouse fibroblasts (L929) were purchased from Shanghai Zeye Biotechnology Co., Ltd. (Shanghai, China).

### 2.2. Oxidation of Cellulose Nanocrystals

Thoroughly desiccated, 2 g of CNC suspension was transferred into a 500 mL beaker, and 3.85 g of NaIO4 was added. The mixture was kept away from light and stirred at 20 °C for varying durations (12, 24, and 48 h), following which 15 mL of ethylene glycol was added to quench the oxidation reaction. The mixture was then dialyzed against distilled water for 9 days.

### 2.3. Functionalization of Carboxymethyl Cellulose Nanofibers

The functionalization of carboxymethylcellulose nanofibers was conducted according to the method outlined by Hudson [26]. Initially, 1 g of dry CMCNF suspension was poured into a 500 mL beaker. Subsequently, dihydrazide adipate (3 g) was added, and the pH was adjusted to 6.8. HOBt (258 mg suspended in a 2-mL DMSO/water (1:1) mixture) and EDC (262 mg in a 2 mL DMSO/water (1:1) mixture) were then added while stirring for 24 h at pH 6.8. The modified compound underwent dialysis in distilled water for 9 days.

### 2.4. Preparation of DACNC/CMC Hydrogels and CMC/ DACNC/ HCNF Hydrogels

Firstly, in Figure 1, 1 g of DACNC obtained from different oxidation times (12 h, 24 h, and 48 h) was placed into a 250 mL beaker; 150 mL of deionized water was added, and the uniform DACNC suspension was obtained by stirring for 30 min at 700 rpm with a magnetic agitator. In addition, 11.25 g of CMC was slowly poured into 250 mL of deionized water, stirred, and added at a speed of 1000 rpm until a uniform yellow solution was formed. Hydrogels (DACNC/CMC-12, DACNC/CMC-24, and DACNC/CMC-48) were obtained by mixing CMC solution and DACNC suspension at a 1:1 volume ratio. For the DACNC/CMC/HCNF hydrogel, 0.6 g, 1.4 g and 1.8 g HCNF were added to the prepared HCNF solution, and HCNF was dispersed into a uniform suspension by ultrasonic cell crushing. Then, the CMC solution containing HCNF suspension and the selected DACNC suspension were mixed at a volume ratio of 1:1 to obtain hydrogels DACNC/CMC/HCNF1, DACNC/CMC/HCNF2, and DACNC/CMC/HCNF3, in which the mass fraction of HCNF was 0.2%, 4%, and 0.6%, respectively.

### 2.5. Verification of Modified Raw Materials and Hydrogels

Fourier transform infrared (FTIR) spectroscopy was used for the infrared characterization of HCNF, DACNC, and hydrogels. Post-modification, the crystal structures of freeze-dried HCNF and DACNC were compared via X-ray diffraction (XRD) analysis.

The crystallinity of the raw material was measured by an X-ray diffractometer. Crystallization index (CrI) according to the Segal equation:(1)$$\mathrm{CrI}\;\left(\%\right)=\frac{I_{200}-I_{am}}{I_{200}}$$
where I_200_ is the diffraction intensity in the crystalline region of cellulose, and I_am_ is the diffraction intensity in the amorphous region of cellulose.

The FTIR spectra of raw materials and hydrogels were analyzed by Fourier transform infrared spectroscopy. Nelson proposed an empirical method for determining the crystallinity of cellulose by infrared spectroscopy, which is expressed by the crystallization index N.O’KI:(2)$$N.O’\mathrm{KI}=\frac{{a1370\;cm}^{-1}}{{a2902\;cm}^{-1}}$$
where ‘*a*’ is the band intensity of the corresponding band in the infrared spectrum. Among them, the 1372 cm^−1^ band belongs to the CH bending vibration, and the 2902 cm^−1^ belongs to the CH and CH_2_ stretching vibrations.

The swelling rate was calculated according to the following equation:(3)$$\mathrm{swelling}\;\mathrm{ratio}\;\left(\%\right)=\frac{W_{t}-W_{0}}{W_{0}}$$
where W_t_ and W_0_ are the hydrogel weight at time *t* of water absorption and the weight of the last test, respectively.

The rheological properties were evaluated, and the morphologies of freeze-dried samples and hydrogels were analyzed using a scanning electron microscope.

### 2.6. Self-Healing Research

The self-healing ability of hydrogels was investigated by cutting them into two parts and then joining them at the break. Observations were made after the samples were left for varying durations at 37 °C. The self-healing ability was quantitatively evaluated using the dynamic oscillation step strain analysis by collecting G’ data at a fixed frequency of 10 rad/s and an alternating strain period of 1% for each 100 s phase of the test.

### 2.7. In Vitro Cytocompatibility Evaluation

The sterilized hydrogel was incubated in a culture medium at an extraction rate of 0.2 g/mL for 72 h. A 200 μL L929 cell suspension was inoculated into a 96-well plate and incubated at 37 °C for 24 h. Subsequently, the culture medium was replaced with a fresh medium containing 50 μL of extract, whereas the control group was maintained without any extract. After incubating for 24, 48, or 72 h at 37 °C, 20 μL of MTT solution was added to the corresponding wells and incubated at 37 °C for 4 h. The culture medium was removed and replaced with 200 μL of dimethyl sulfoxide in each well. The optical density was read at 490 nm. A live/dead test was performed using a live/dead kit. L929 cells were cultured at 37 °C for 24, 48, or 72 h, then washed twice with phosphate buffer and stained according to the instructions of the live/dead kit.

## 3. Results and Discussion

### 3.1. Formation of CMC/DACNC/HCNF Hydrogels

The FTIR spectra of CNF obtained under different oxidation time conditions are shown in Figure 2a. In general, CNC samples have absorption peaks at 3400 cm^−1^, 1429 cm^−1^, 1370 cm^−1^, 1322 cm^−1^, and 897 cm^−1^. These correspond to O–H tensile vibration, C–H shear vibration, C–H bending vibration, O–H in-plane bending vibration, and C–H deformation vibration, respectively, which are the typical characteristic peaks of cellulose. It can be seen that the prolongation of oxidation time will not change the main structure of cellulose. In addition, a new absorption peak in the spectrum of DACNC is revealed at 1729 cm^−1^, which signifies the presence of an aldehyde group. The successful modification of CNC is confirmed by the appearance of the aldehyde group peak in the oxidized DACNC spectrum. As oxidation time increased, the aldehyde group content increased, and the content of aldehyde was quantified by the corresponding oxime nitrogen content, as reflected in aldehyde contents of 0.73, 1.09, and 1.61 mmol·g^−1^.

The infrared spectrum of HCNF is shown in Figure 2b. Both CMCNF and HCNF have characteristic peaks of cellulose. The absorption peaks at 3400 cm^−1^, 1430 cm^−1^, 1370 cm^−1^, 1322 cm^−1^, and 897 cm^−1^ correspond to O–H tensile vibration, C–H shear vibration, C–H bending vibration, O–H in-plane bending vibration, and C–H deformation vibration, respectively. More importantly, it presents a new peak at 3541 cm^−1^, attributable to NH stretching vibration, the new absorption peak at 1705 cm^−1^ is the C=O stretching vibration in the characteristic peak of hydrazide. In addition, the symmetrical stretching vibration peak of the carboxylic acid group of the carboxymethyl structure at 1601 cm^−1^ in the infrared spectrum of CMCNF also shifted, which proved that the amide grafting was successful. X-ray photoelectron spectroscopy (XPS), which was used to evaluate the constituent functional group elements of HCNF, shows the appearance of an N1s peak in HCNF, a feature not present in CMCNF (Figure 2c). The N1s curve, shown in Figure 2f, matches the binding energies of 400.1 and 402.1, which are attributed to –NH and –N=, respectively, corroborating the FTIR spectroscopy results.

The crystallization properties of the compounds were analyzed via an X-ray diffractometer. Figure 2d shows typical cellulose peaks for the cellulose sample, with primary peaks located at 14.9°, 16.5°, and 22.7°, corresponding to (1–10), (110), and (200), respectively. N.O’KI was 1.31, 1.27, 1.21, and 1.17, respectively, which was consistent with that of CrI. Despite the decrease in DACNC crystallinity with reaction time, the diffraction peak angle remains largely unchanged, suggesting that the fiber maintains the nanocrystalline structure of cellulose.

Figure 2e shows that the hydrogel shows a new absorption peak at 1670 cm^−1^, which belongs to the stretching vibration of the imide bond or the stretching vibration peak of the hydrazide bond. And the disappearance of the NH stretching vibration characteristic peak and aldehyde group absorption peak in the infrared spectrum of the hydrogel. The intensity of the 3402 cm^−1^ absorption peak also decreases, indicating cross-linking between the aldehyde group and amino and amide groups.

Scanning electron microscopy (SEM) was employed to characterize the morphologies of the raw materials and hydrogels (Figure 3). The SEM image of DACNC (Figure 3a) reveals fiber cluster binding, while HCNF (Figure 3b) exhibits a closely intertwined fiber porous structure. In contrast, CMC (Figure 3c) displays a smooth surface. The hydrogel (Figure 3d) demonstrates an irregular, dense, porous structure, indicating successful HCNF cross-linking with the Schiff base, enhancing the hydrogel’s network structure. The porous structure of hydrogels facilitates cell growth, promotes wound healing, and allows the spread of gases, nutrients, and waste [27].

### 3.2. Fluid Absorption Capacity

Figure 4 presents the properties of the DACNC/CMC hydrogel, focusing on its storage modulus (G’) and loss modulus (G’’) at different speeds, water loss ratios, and swelling ratios. As shown in Figure 4a, the hydrogel’s G’ value, representing the energy storage modulus, increases with the degree of aldehyde substitution in DACNC; this is attributed to the increased interaction between the aldehydes and amino groups in CMC, leading to a more closely interconnected network capable of better pressure resistance [14,28]. Based on these results, DACNC−48 was selected for the follow-up experiments, and the hydrogels containing DACNC in subsequent experiments were also DACNC-48.

Figure 4b shows that the storage modulus of the hydrogels decreases with increasing HCNF content, which can be attributed to the excessive tendency of HCNF to aggregate, reducing their dispersion ability and leading to a decrease in the cross-linking point and degree of the hydrogel cross-linking. The HCNF1 and HCNF2 contents in the follow-up experiments were 0.2% and 0.4%, respectively.

The ability of the hydrogel to absorb wound exudate and maintain moisture on the wound surface is a crucial factor for wound healing applications. The hydrogel’s swelling ability was evaluated by immersing it in a PBS buffer (pH 7.4) for different periods. As depicted in Figure 4c, the hydrogel maintained its structural integrity in water after seven days of testing, whereas traditional self-healing hydrogels synthesized by soft chains did not. The swelling rate of hydrogel in equilibrium is as high as 381%, which is higher than that of cellulose-based hydrogel (300.7%) [29]. The preservation of the hydrogel structure and high water absorption are attributed to two main factors. First, the network structure synthesized by cross-linking DACNC with CMC and HCNF reduced the movable space and chain fluidity in the hydrogel, inhibiting water infiltration, adhesion, and absorption while maintaining a firm hydrogel structure. Second, adding the HCNF limited the fluidity of the soft CMC chain in the hydrogel network but led to a decrease in water absorption at higher HCNF content. Furthermore, HCNF strengthened the hydrogen bonding with water while reducing the chain fluidity, thereby reducing water loss (Figure 4d).

### 3.3. Mechanical Strength

Hydrogels for the test were prepared in a mold, and the tensile tests of DACNC/CMC and DACNC/CMC/HCNF hydrogels were carried out. The tensile strength of DACNC/CMC/HCNF2 increased significantly, and the maximum tensile strength of 164.2 kPa was higher than that of nano-cellulose chitosan dressing [30]; the tensile strength of DACNC/CMC/ HCNF1 was 135.1 KPa, whereas that of DACNC/CMC was 100.9 KPa.

Figure 5d presents the compressive strength of hydrogels with different HCNF contents. The compressive strength of all hydrogel samples increases with compressive strain, peaking at 50% strain. With the increase in HCNF content in hydrogel, the maximum stress of DACNC/CMC/HCNF hydrogel increased from 17.8 kPa to 41.9 kPa, which was higher than that of gelatin cellulose hydrogel [30], reflecting the notable enhancement effect of HCNF. As shown in Figure 5e, DACNC/CMC/HCNF hydrogels essentially align after 10 compression cycles, demonstrating the excellent compression properties of the hydrogels.

In Figure 5c, the G’ value of the DACNC/CMC hydrogel quickly surpasses its G’’ value, and the G’ values of other hydrogels surpass the corresponding G’’ values, suggesting that the hydrogels possess a cross-linking structure and are stable. The introduction of HCNF significantly increases the viscosity, and the G’ and G’’ values of the hydrogel reach 557 and 617 Pa, respectively, indicating an improvement in mechanical properties.

The study utilizes continuous step strain measurement, wherein shear strain is alternately applied (Figure 5f). After switching from 200% high strain to 1% low strain, G’ and G’’ rapidly return to their original values without a significant decrease; this can be attributed to the reconstruction of hydrogel networks by dynamic imide, hydrazone, and hydrogen bonds at low strain.

### 3.4. Self-Healing Performance

The self-healing ability of materials can prolong their service life. In this work, we prepared a hydrogel with self-healing ability via a reversible Schiff base reaction. The DACNC/CMC/HCNF hydrogel was cut into two parts and allowed to come into contact without any external force for 5 min. First, we can visually see the hydrogel fragments self-healing together, demonstrating their self-healing ability. Furthermore, after 12 h, the hydrogel could be compressed without any sign of rupture (Figure 6f). As shown in Figure 6a, consistent with the infrared spectrum of the hydrogel synthesis, there is a new absorption peak at 1670 cm^−1^, which belongs to the stretching vibration peak of the imide bond or the stretching vibration peak of the hydrazide bond.

Figure 6b,c shows the tensile and compressive capabilities of the DACNC/CMC/HCNF2 hydrogel before and after healing at 37 °C. It can be clearly seen that as the healing time increases, the tensile and compressive capabilities are gradually restored because the active groups in the hydrogel are re-crosslinking to restore the gel structure. Figure 6d and e illustrates the G’ value and corresponding healing efficiency data of DACNC/CMC/HCNF2 hydrogel before and after healing at 37 °C. The G’ value after 24 h was nearly identical to that of the original hydrogel, and the healing efficiency reached a remarkable 90%. The results indicate that the dynamic chemical and hydrogen bonds are destroyed when the hydrogel is divided into two parts, and the free active groups are exposed at the broken interface. These reactive groups demonstrate a tendency toward rearrangement. Thus, the dynamic imine, hydrazone, and hydrogen bonds are quickly combined and recombined after surface re-contact, enabling the network to repair itself quickly and effectively.

The healing process of the hydrogel was observed using an optical microscope (Figure 6g). For DACNC/CMC/HCNF2 hydrogels, the healing speed is faster than that for other hydrogels, indicating that the healing speeds of DACNC and CMC increase in the same proportion with the increase in HCNF and NH2/-CHO molar ratios. SEM images show that the hydrogel healed without gaps after 12 h (Figure 6h).

### 3.5. Antibacterial Activity

The hydrogel under investigation contains chitosan, a well-known biopolymer with inherent antibacterial properties. To investigate the antibacterial activity of the hydrogel, Figure 7 displays its performance against two common bacteria, Escherichia coli and Staphylococcus aureus. The results show that the hydrogel can effectively inhibit bacterial growth, particularly against Escherichia coli, with antibacterial activity exceeding 50% after three days. However, the antibacterial activity decreases as the HCNF content increases; this is likely due to the excessive binding of HCNF to CMC via hydrogen bonding, which limits the exposure of amino groups in CMC and consequently reduces the antibacterial activity.

### 3.6. In Vitro Cytotoxicity

Figure 8a presents the results of the indirect cytotoxicity test. Compared with the negative control, all hydrogel formulations except for DACNC/CMC and DACNC/CMC/HCNF1 showed higher cell viability after the first day. This indicates that the hydrogel does not exhibit toxicity toward L929 cells. Furthermore, live/dead assays were performed, with results shown in Figure 7b. After culturing with the hydrogel extract, the viable cells, stained green, demonstrated a proliferation trend similar to that of the negative control group. These results suggest that the hydrogel has excellent biocompatibility. Moreover, producing these hydrogels does not involve any potentially harmful foreign cross-linking agents.

## 4. Conclusions

In this study, we combined a flexible CMC chain with an aldehyde group-functionalized CNC and hydrazide CNF to form a DACNC/CMC/HCNF hydrogel. This hydrogel demonstrated exceptional self-healing capability, with a healing efficiency of 90% and an equilibrium swelling rate of 381%. Furthermore, HCNF significantly improved the tensile and compression elasticity coefficients, reaching 164.2 and 41.9 KPa, respectively. The exceptional self-healing properties of this hydrogel can be attributed to the synergistic effect of dynamic imine, hydrazone bonds, and hydrogen bonds. Additionally, the hydrogel displayed excellent biocompatibility and effectively supported cell growth as an extracellular matrix. The cell viability after culture was also higher than that of the control group. Considering these superior properties, the newly developed hydrogel is a promising candidate for skin wound repair applications.

## Figures and Tables

**Figure 1 polymers-15-03389-f001:**
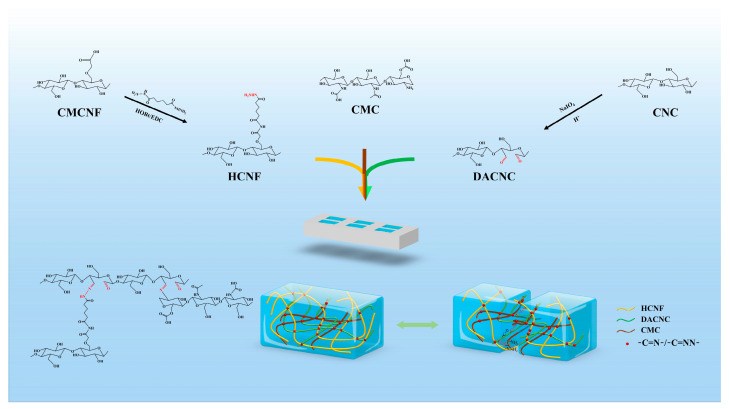
Schematic representation of the synthesized DACNC/CMC/HCNF.

**Figure 2 polymers-15-03389-f002:**
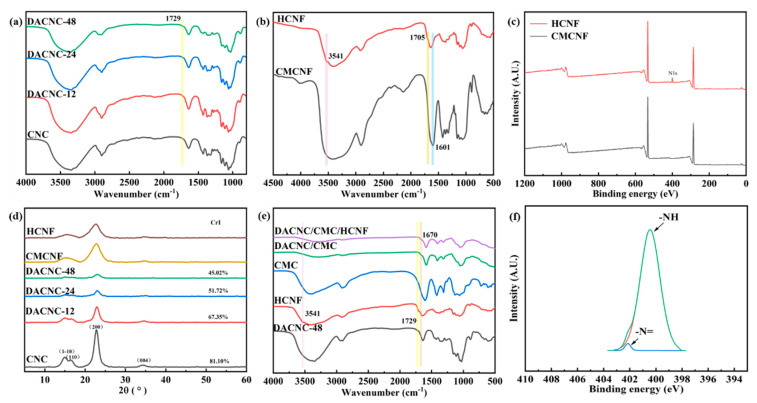
(**a**) FT–IR spectra of CNC, DACNC-12:48; (b) FT–IR spectra of HCNF, CMCNF, DACNC/CMC, and DACNC/CMC/HCNF; XPS of (**c**) HCNF and (**f**) N1s; (**d**) XRD curves of CNC, DACNC-12, DACNC-24, DACNC-48, HCNF, and CMCNF; (**e**) FT–IR spectra of HCNF, CMCNF, DACNC/CMC, and DACNC/CMC/HCNF.

**Figure 3 polymers-15-03389-f003:**
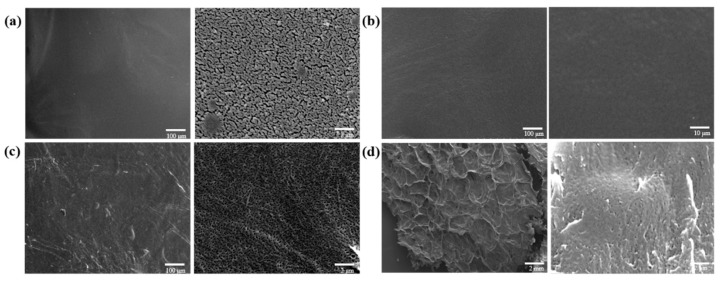
SEM images of (**a**) DACNC, (**b**) CMC, (**c**) HCNF, and (**d**) the DACNC/CMC/HCNF2 hydrogel.

**Figure 4 polymers-15-03389-f004:**
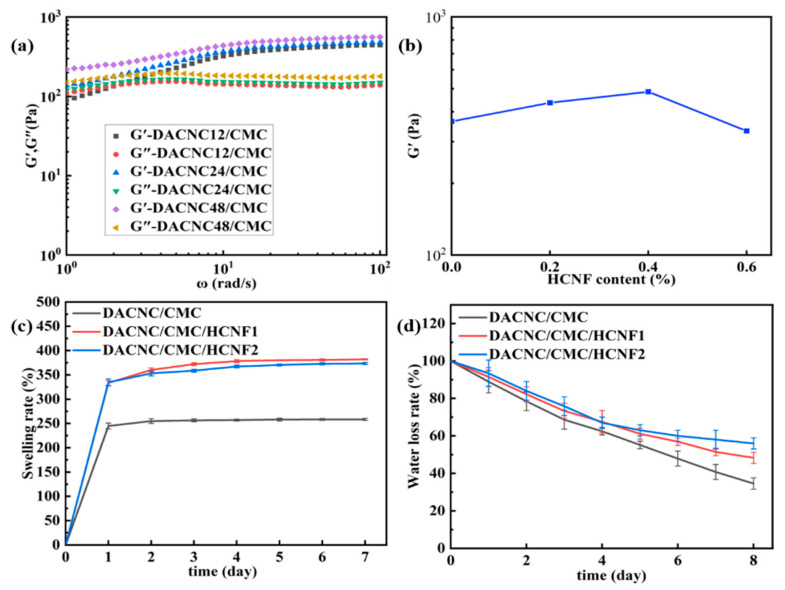
(**a**) Storage modulus (G’) and loss modulus (G’’) of the DACNC/CMC hydrogel at different speeds. (**b**) Storage modulus (G’) of the hydrogel at 10 rad/s. Water (**c**) swelling ratios and (**d**) loss ratios.

**Figure 5 polymers-15-03389-f005:**
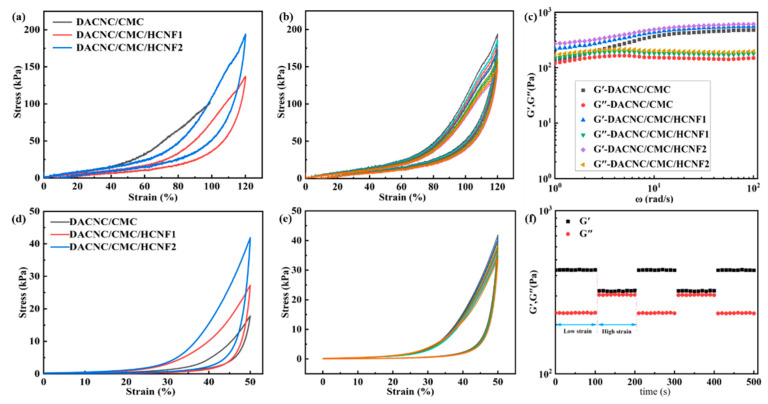
(**a**) Tensile properties of DACNC/CMC, DACNC/CMC/HCNF1 and DACNC/CMC/HCNF2 hydrogels. (**b**) The 10 stretching cycles of DACNC/CMC/HCNF hydrogel. (**c**) Frequency sweeps of the DACNC/CMC, DACNC/CMC/HCNF1, and DACNC/CMC/HCNF2 hydrogels. (**d**) Compression properties of DACNC/CMC, DACNC/CMC/HCNF1, and DACNC/CMC/HCNF2 hydrogels. (**e**) The 10 compression cycles of DACNC/CMC/HCNF2 hydrogels. (**f**) Alternating strain scanning of DACNC/CMC/HCNF2 hydrogel at 10 rad s^−1^ frequency.

**Figure 6 polymers-15-03389-f006:**
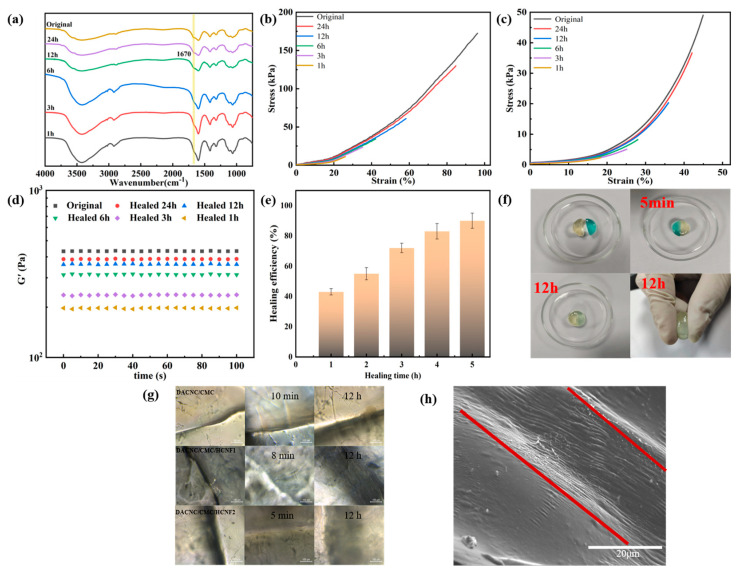
(**a**) FT−IR spectra of DACNC/CMC/HCNF2 hydrogel self-healing at different times. (**b**) Tensile properties of DACNC/CMC/HCNF2 hydrogels at different times after healing. (**c**) Compression properties of DACNC/CMC/HCNF2 hydrogels at different times after healing. (**d**) G’ values before and after healing. (**e**) Healing efficiency test. (**f**) Self-healing pictures of DACNC/CMC/HCNF2 hydrogel. (**g**) Microscopy photographs of healing processes after failure induced by cutting. (**h**) SEM image of the DACNC/CMC/HCNF2 after 12 h of healing.

**Figure 7 polymers-15-03389-f007:**
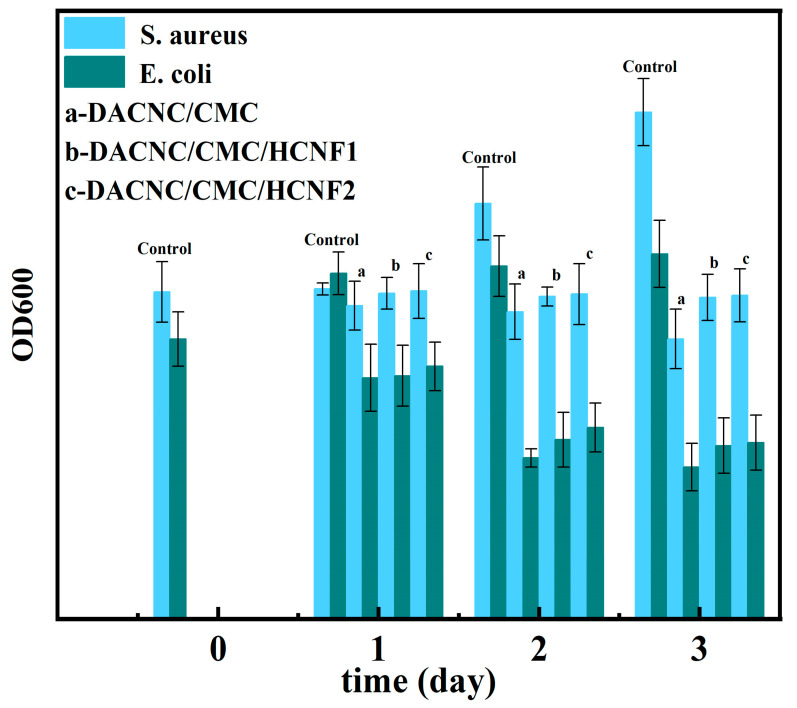
The OD600 value of bacterial culture.

**Figure 8 polymers-15-03389-f008:**
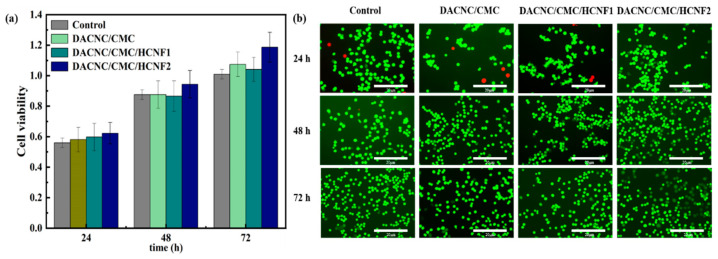
MTT and live/dead assays. (**a**) Cell viability; (**b**) staining photographs of cell survival.

## Data Availability

All data generated or analyzed during this study are included in this published article.

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
