# Peer review of "Wound Dressing Double-Crosslinked Quick Self-Healing Hydrogel Based on Carboxymethyl Chitosan and Modified Nanocellulose"

_polymers, 2023, doi:10.3390/polym15163389_

Round 1

Reviewer 1 Report

The focus of the manuscript is on developing double crosslinked quick self-healing hydrogel dressing based on carboxymethyl chitosan and modified nanoscelluloses.

The paper is well written, however, several points should be cleared before acceptance to this journal.

The names of the hydrogels are very confusing.  For example, in conclusion DACNC/CMC/HCNF in Figure 6.  In Figure 6a, it only showed one hydrogel, however, the caption showed DACNC/CMC and DACNC/CMC/HCNF hydrogel.  But not showing it is DACNC/CMC/HCNF-1 or DACNC/CMC/HCNF-2.  Also it lacks the information of DACNC/CMC.   Please check it.

Also, for Figure 6f the anti-bacterial study, it is too small to read.  Please put it in a individual figure.

The English part looks OK, however, it might require more professional editing.

Author Response

The following is an item-by-item answer to the opinions of the reviewers:

  1. The names of the hydrogels are very confusing. For example, in conclusion DACNC/CMC/HCNF in Figure 6.  In Figure 6a, it only showed one hydrogel, however, the caption showed DACNC/CMC and DACNC/CMC/HCNF hydrogel.  But not showing it is DACNC/CMC/HCNF-1 or DACNC/CMC/HCNF-2. Also it lacks the information of DACNC/CMC.   Please check it. Also, for Figure 6f the anti-bacterial study, it is too small to read.  Please put it in a individual figure.

Thanks to the patient reviewers, the name of the hydrogel has been revised in the article. Pictures of antibacterial research have also been released separately. The revised part is highlighted in yellow.

  1. -section 2.4 - the preparation of the hydrogels must be presented for all samples showed in Fig. 2.

Thanks to the reviewers for their valuable advice, the preparation of the hydrogel has been written in detail.

3. The English part looks OK, however, it might require more professional editing.

Thank you for your valuable advice, after adjusting a pair of sentences.

Reviewer 2 Report

The paper "Double Crosslinked Quick Self-Healing Hydrogel Wound Dressing Based on Carboxymethyl Chitosan and Modified Nanocelluloses" reports on the preparation, characterization and investigation of self-healing behavior of chitosan and cellulose derivatives based hydrogels. The idea of this manuscript is well presented, but some improvements are needed as follows:
-the title should be corrected as: "Wound Dressing Double Crosslinked Quick Self-Healing Hydrogel Based on Carboxymethyl Chitosan and Modified Nanocellulose"

- the abstract should be rephrased to be better understood;

-section 2.4 - the preparation of the hydrogels must be presented for all samples showed in Fig. 2.

-some peaks must be shown in Fig. 2.

-FT-IR spectra must be explained, the formation of the new bonds: imide and Schiff base must be highlighted. The crystallinity of the samples also can be evaluated based on FT-IR spectra and compared with XRD data.

-Fig, 3 - the morphology of the gels is too compact, the SEM are registered on dried samples?

-Some comparisons with literature data must be added in section 3.2 and 3.3 on swelling data, G' and G'' values, and mechanical data.

-section 3.4-the healing also must be evidenced by FT-IR spectra and mechanical data.

Some Conclusions (lines 310-311) must be supported by the experimental data not by assumptions.

My recommendation is Major revision for this manuscript.

Author Response

The following is an item-by-item answer to the opinions of the reviewers:

  1. -some peaks must be shown in Fig. 2.

The picture in figure 2 has been modified and the main peak has been marked.

  1. -FT-IR spectra must be explained, the formation of the new bonds: imide and Schiff base must be highlighted. The crystallinity of the samples also can be evaluated based on FT-IR spectra and compared with XRD data.

The formation of hydrogels and the formation of new bonds have been explained. The revised part is highlighted in yellow.

  1. -Fig, 3 - the morphology of the gels is too compact, the SEM are registered on dried samples?

With regard to this problem, the sample is freeze-dried and contains no moisture.

  1. -Some comparisons with literature data must be added in section 3.2 and 3.3 on swelling data, G' and G'' values, and mechanical data.

Thanks to the reviewers for their valuable comments on the structure of the article, comparative data have been added. The revised part is highlighted in yellow.

  1. -section 3.4-the healing also must be evidenced by FT-IR spectra and mechanical data.

The experiment has been supplemented and analyzed in accordance with the opinions. The revised part is highlighted in yellow.

  1. Some Conclusions (lines 310-311) must be supported by the experimental data not by assumptions.

Thank you for reading it by yourself and pointing out the wrong data. There is a sentence in writing that has not been deleted completely. The revised part is highlighted in yellow.

All the best,

Round 2

Reviewer 2 Report

The authors have improved their manuscript according to the suggestions and can be accepted in this form.